

**A New Way to Estimate Maximum Power from Wind Turbines: Linking**
**Newtonian with Action Mechanics**
Ivan R. Kennedy,[1,3,*] Migdat Hodzic,[2] Angus N. Crossan,[3] Nikolas Crossan,[3] Niranjan Acharige,[1]
John W. Runcie[1]
[1]School of Life and Environmental Sciences, Institute of Agriculture, University of Sydney, NSW
2006, Australia
[2]Faculty of Information Technologies, The Dzemal Bijedic University of Mostar, Mostar 880000
Bosnia and Herzegovina
[3]Quick Test Technologies, c/- Institute of Agriculture, University of Sydney, NSW 2006, Australia
Corresponding author: ivan.kennedy@sydney.edu.au
**Abstract**
A more accurate way to calculate power output from wind turbines based on fundamental Newtonian
mechanics is proposed for testing. This contrasts with current methods regarded as governed by flows
of kinetic energy through an area swept by rotating airfoils. Action mechanics measures torques caused
by conservation of momentum of impulsive air streams on rotor surfaces at differing radii. We integrate
the windward torque using inputs of rotor dimensions, the angle of incidence and strength of wind
impulses on the blade surfaces. A reverse torque in the plane of rotation is estimated as radial impulses
from the blade's rotation. Net torque is converted to power by the angular velocity of the turbine rotors.
A matter of concern is significant heat production by wind turbines, partly from leeward reactions but
mainly from turbulent release of vortical energy. Use of wind farms as sources of renewable energy
may need better practice, minimizing environmental impacts guided by this hypothesis.
### 1. Introduction
Current models of wind turbine function use aerodynamic principles derived largely from airfoil and
propeller theories. The Rankine-Froude momentum and actuator disk models were developed in the
19th century with Betz and later Glauert (1935) providing refinements related to wind turbine
efficiency, including more recent developments (Sorensen, 2015). These models may include losses
from the axial motion of the air induced by rotors, in marked contrast to the radial action model
proposed here that assumes turbine blades generate power while rotating into undisturbed air,
independent of the down-wind wake.
A detailed explanation of the more recent blade element momentum theory (BEM) need not be given
here. In brief summary, BEM leads to an inexact expression for power output ($P$) according to the
following equations (1) and (2) as a function of the cube of wind velocity ($v$), air density ($\rho$), the area
swept ($A_D$) by the rotor blades with diameter $D$ and a specific axial induction factor ($a$) related to
changes in angular momentum of the air flow.
$$P = [0.59v^3\rho A_D]/2 \tag{1}$$





$$P \;=\; 2a(1 - a)^2 v^3 \rho A_D \qquad\qquad (2)$$

This enables power extraction for a system that includes a rotating wake, claimed to give a maximum power consistent with the Betz limit for power from the kinetic energy of 0.593. Taken with other inefficiencies, a power output of about 30% of the theoretical maximum possible from kinetic energy is found in practice. Forces from air flow over an airfoil responding to angular dimensions of the blade are decomposed into lift and drag, normal and tangential to the apparent wind speed. This enables estimates of forces rotating the turbine and those just bending the rotor to be separated, taking the axial factor ($a$) of equation (2) into account to estimate torques. However, we will point to flaws in this model, caused by mismatching the interception of wind energy by the blades and the use of a model of the inertial power of wind we judged as inferior (Kennedy and Hodzic, 2021a).

We have defined with action mechanics statistical variation in radial separation and temperature of molecules undergoing impulsive collisions, allowing estimation with the accuracy needed to calculate the action and entropy of atmospheric gases (Kennedy et al., 2019). In our subsequent revision of the Carnot cycle (Kennedy and Hodzic, 2021a, 2021b), such molecular action states were found to define the density of quanta that establishes a Gibbs field needed to sustain the molecular kinetic pressure responsible for work processes. Molecular kinetic energy alone was insufficient to explain these processes. This complementary notion of the significance of the Gibbs action field challenges the widespread assumption that heat is merely molecules in motion.

We speculated during 2021 that similar complementary action processes might explain the inertial wind pressures acting on the blades of turbines. The scientific question we ask is whether this developing theory can be a more accurate way to estimate power output from wind turbines by using a similar coupling of Newtonian to action mechanics (Kennedy et al., 2021). Given Isaac Newton's youthful interest in the causal theory of windmills shown by his recently discovered inscriptions on the farmhouse walls at Woolsthorpe, this possibility seems apt. The purpose of this paper is to test this hypothesis from its predictions, allowing the exercise of Ockham's razor to judge its success.

**1.1 *Radial action theory***
In the radial-action model for estimating wind power the details of blade aerodynamics need only be considered later as refinements. Similar to Carnot's purpose in defining cycle for heat engines (Kennedy and Hodzic, 2021a) the radial action cycle for wind turbines should be considered as an ideal estimate of the maximum possible motive power. Inefficiencies from friction or other causes will not be dealt here. Radial action mechanics should apply to rotor blades of any shape, while the fundamental differences in geometry and the torques generated by the windward [$T_w$] and leeward surface [$T_b$] must be respected. Figure 1 models the torques generated on the blade surface areas, allowing the maximum power to be estimated as a function of windspeed, its angle of incidence and the actions and reactions in the blade surface material, controlled by the blade length ($L$ or $R$). chord width ($C$) and the tip-speed ratio [$L\Omega/v = \lambda$] of rotor tip speed [$L\Omega$] compared to wind speed [$v$].



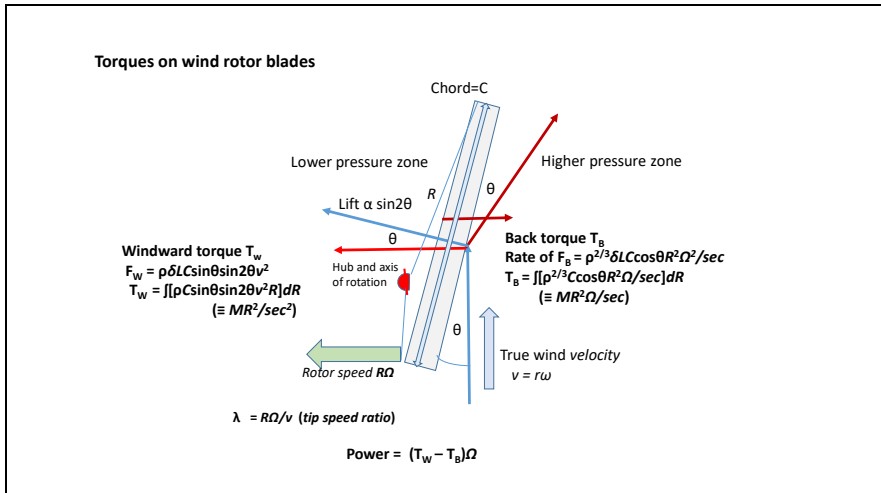

**Figure 1.** Windward ($T_w$) and leeward ($T_b$) back torques developed on a rectangular rotor blade. Equations were generated from trials using a numerical program using dimensions shown. The theory provides a theoretical maximum of power for a given angle of wind incidence, obtained by the difference between torques $T_w$ and $T_b$, occurring at about θ=60°. Note the different dimensions employed for the density factor, explained in the text.

The main results found from our analysis of the impulsive action on the windward surface of the blade (Figs. 1, 2) follow.

(i)     Impulses [$\delta mv=\delta mr\omega$] generated by material particles on elastic rotor surfaces as envisaged by Newton's second law experimental approach to conservation of momentum power the turbine's rotation at the hub, if free to do so. The air particles impact on trajectories imposing the inertia of the wind velocity on their far greater microscopic velocities, with mean free paths of the order of picometers. Action impulses [$\delta mvR$, J.sec] with $R$ the radial dimension to the hub, reflect the momentum of air trajectories from the surface. No detailed consideration needs to be given to the individual trajectories of the air molecules comprising wind, given that the transfer of momentum is collective. The rate of impulsive action [$T_w = \Sigma \delta mvR/\delta t$, J or N.m] provides the magnitude of the windward torque exerted.

(ii)    The angle of incidence θ is also the angle of reflection from a flat surface (Fig. 1), giving a total deviation angle of 2θ for the momentum. The decreased forward velocity is the source of effective lift normal to the flat surface, determined by this angular deviation.

(iii)   A turning moment is exerted within the blade material in the plane of freedom of rotation of the turbine, at an angle normal to the direction of wind incidence. The magnitude of the turning moment [$Mv\sin2\theta$] and its cause is illustrated in Figure 2.

(iv)    This application of Newton's second law requires that the true wind direction be considered to estimate the reaction on the turbine blade. The apparent change of wind direction caused by the rotation of the rotors, critical in BEM theory, is irrelevant in the radial action model. The current blade momentum theory using aerofoil theory as an analogue assumes lift normal to the blade and drag in the same direction as the blade; taking account of axial air motion downwind is an unnecessary confusion of cause and effect for impulsive radial action.



(v)     Two kinds of impulsive force on the blades need to be considered. The first is uniform with
length along the blade from reflected windward impulses on the blades' surfaces, giving
the torque generating the rotation; the second is a reaction torque also variable with blade
radius $R$ from impacts by the rear of the advancing blades on air molecules, tangential to
the direction of rotation (Fig. 1). At tip-speed ratios greater than one, the back torque
involves impulses of greater magnitude than ever seen in airfoils unless rapidly gaining
altitude. As a result for most of the blade, except that adjacent to the hub, no drag force
operating in the wind direction can be caused by turbulence on the downwind surface given
the normal reaction from air molecules to the rear of the blade as Newton's marbles; drag
has little or no analogue in wind turbines (Figs. 1, 2). If the aerofoils of aircraft are
considered the same as the blades of wind turbines, the aircraft should be rotating around
its longitudinal axis as it is impelled forward by propellers or the thrust of jets. It is
suggested this discrepancy could make BEM theory a flawed approach.

**Figure 2.** Generation of turning impulse ($V\sin2\theta$) from the surface normal to the wind direction is shown for
two angles of incidence. Highly elastic action causes stresses and strain in reaction that intensify chemical
potential in rotor material, eased by rotation action exerted as a degree of freedom. For the perfect elasticity of
1.0, the turning moment imparted to the rotor balances that of the reflected momentum in the opposite direction.

**1.2 *Analyzing of the rotor-turning moment from wind pressure***
Whatever paths the individual air molecules take in flow near the blade surface, dictated as laminar
while the Reynolds number remains small, it is a fundamental principle of Newtonian mechanics that
the linear momentum at a given radial action is conserved. By Newton's experimental law the impact
of elastic bodies for oblique collisions on smooth surfaces will be reflected by the same angle for
coefficients of restitution of 1.0. However, part of the moment exerted by wind particles may be
extinguished if absorbed as thrust acting to push the wind tower and blades in a direction unable to
rotate the rotor on its axis. For smooth surfaces there is no force parallel to the surface and the
component of the particle velocity in the direction of motion is shown in Figures 1 and 2 as fully
conserved with the angle of reflection equal to the angle of incidence.



For oblique impacts in the range 0-90 degrees, asymmetric compression of blade material at the surface
are generated as an oscillating function, dependent on the elasticity and density of the blade material.
If the blade surface remains clean and elastic, this variation in stress as reactive pressure produces
strains distorting the windward surface, varying the chemical potential in the compressed zones as a
function of the angular deviation of the reactions. As a relationship between elastic stress and strain,
this reaction can be described as a function of Young's modulus ($E$) for the rotor surface material, with
surface stress σ or uniaxial force ($m\omega^2$) and strain ε equal to the distortion $\delta l/l$.

$$E = \sigma/\varepsilon \tag{3}$$

For the blade material distal to the point of wind reflection, the physical reaction to the radial impulses
is distributed in an arc of (90+θ) degrees while for the proximal reaction the arc for compression is
(90-θ). The arc difference being 2θ, the turning moment per molecule on the rotor's axis is proposed
to vary with $mv\sin2\theta$, thus balancing action and reaction. At 90 degrees or π/2 radians, this function
becomes zero with any turning moment now symmetrical totally devoted to bending rather than turning
the blade on its rotational axis at the hub. Obviously, this analysis can only be applied to compressions
on the windward side of the rotor blades. The other factors determining the windward torque are the
density of air (ρ), the chord width ($C$) and sinθ, determining the volume and the mass of air impacting
the blade per second. This represents the instantaneous magnitude of mass impacting per second for
the area normal to the wind flux. When integrated with respect to the radius ($R$) over the entire length
of the blade ($L$) the cumulative torque [$T_w$] exerted at the hub is given in equation (4).

$$T\text{w} \;=\; \int_0^L [\rho\delta LC\sin\theta\sin2\theta\; v^2 R]dR = \int_0^L [M\sin2\theta\; vR/sec]dR \quad [ML^2T^{-2}, \text{J or N.m}] \tag{4}$$

This equation comprises factors for the 3-dimensional density of air (ρ), the area of the blade at $R$
($\delta LC\sin\theta$), the momentum per sec [$\rho\delta LC\sin\theta v = Mv$] made normal to the wind by sinθ, the extent of
lateral reaction, thus [$Mv\sin2\theta$]. Numerically, the square of the wind velocity ($v^2$) is involved, once to
estimate the mass of air impacting the blade per second and second to establish the magnitude of action
impulse per second proportional to the variation in action ($\delta mvR$) per molecule. This is considered as
involving a rectangular blade in Figure 1 but different versions of the blade area at any radius can be
estimated from variations in the chord width ($C$) as a function of the radius to the hub ($R$). The 3-
dimensional density is regarded as a thermodynamic function, given that wind is a cooperative action
with its radial inertia involving not just the kinetic energy of molecules striking the rotors on the
windward surface, but with vortical energy and its resulting chemical potential. This vortical energy
unique to vortexes is explained in Results and in Discussion.

### 1.3 *Leeward torque of rotor blades*

It is said that a youthful Newton while designing his flour windmill estimated wind force by the
difference between the distances he could leap with and against the wind. This image is consistent with
our model of the impact of the inertia of the blade on that of air. In contrast to the windward torque



$[T_w]$ proportional to radius, in which the wind factor $v^2$ applies uniformly with radius $R$ over the rotor
blade from hub to tip, the back torque $(T_b)$ varies with the square of the radius, caused by the variable
rate of impacts on the air behind the blade variable with $R$ during rotation. This variation is illustrated
in Figure 2. Given that the speed of rotation $[R\Omega, \text{m sec}^{-1}]$ determines both the instantaneous mass of
air impacted per second as well at the radial momentum of these impacts, the specific action integral
required is of radius squared $[R^2\Omega]$. An initial run of the model using $R^2\Omega^2$ as a factor was found to
produce a function of power rather than torque. This justified integrating the specific action per radian
$(R^2\Omega)$ instead of the energy per unit mass $[R^2\Omega^2]$. In effect, a variable inertial force along the blade is
integrated with respect to velocity to provide the correct rate of impact with mass.
An important difference between windward and leeward impulses with air molecules lies in the
irreversible nature of impacts from the blade on air molecules. While windward impulses may be
considered as a balancing of forces from the wind on the blade reaction, the leeward impacts generally
exceed the wind speed except near the hub and cooperative resistance from air at the rear of the blade
is much diminished and is ignored. As a result, the density of air molecules is effectively exerted from
2-dimensional action impulses exerted as a series of minute slices of air, varying with radius. If the
number density of molecules in a cubic meter is taken as proportional to $n^3$ then $n^2$ must represent the
density of molecules the blade encounters as a continuous process. Taking the density as having a
fractional exponent, the factor required should be $\rho^{2/3}$ or 1.145 rather than 1.225. By such a choice the
correct physical dimensions to describe the rate of transfer of momentum from the blade to air
molecules, integrated with respect to $R,$ obtains the rate of impulsive action or reactive torque. The
inertial matter $(MR)$ impacted per second expresses the action function $MR^2\Omega$ rather than momentum
$MR\Omega$, with decreasing orthogonality of impulses on shorter radii. To obtain the reverse torque,
equation (5) must be integrated
$T_b \ = \ \int[\rho^{2/3}Ccos\theta R^2\Omega/\sec]dR \ = \ \int[MR^2\Omega/\sec]dR$     $[ML^2T^{-2}, \text{Nm}](\text{kg.m}^2 \text{per sec}^2)$   (5)
The surface area swept aside by the blade for each 1 meter segment of the length $L$ is $C\cos\theta$, given
that the radius is varied to estimate torque for each decrease in the length of the blade. So the
momentum generated in each second at each radius is equal to the volume swept aside per
second $[C\cos\theta R\Omega$ x density $\rho/\sec = MR\Omega/\sec]$. Expressed as an action impulse depending on the
radius, that gives action per sec or torque $[\rho CCos\theta R^2\Omega/\sec]$ or $[MR^2\Omega/\sec, ML^2T^{-2}]$. The longer the
radius, the more orthogonal the impulse and the effectiveness of the action impact $[mrv, \text{J.sec}]$.
Subtracted from the windward torque exerted on the front of the rotor blade to obtain the net torque
on the rotor, then multiplied by the number of blades and by the angular frequency $\Omega$, the net power $P$
can be obtained.  Both torque equations can be derived with a constant value for any configuration of
rotor operation and then integrated in a standard formula for $R$ and $R^2$ respectively [integrals of $L^2/2$
and $L^3/3$] for accurate outputs, assuming ideal conditions. Taking the derivative of factors such as
angle of incidence, tip-speed ratio $(\lambda)$ and rotor length $(L)$ with respect to power allows optimization
of each of these factors.  This should allow ease of control of these factors in wind turbine operation.
Optimum tip speed ratios are usually in the range of 3 to 10 with optimum length a function of wind
speed. Then turbine power $(P)$ can be estimated by the difference of windward and leeward torques
multiplied by the angular frequency $(\Omega)$ of rotation.



$$P = [T_\text{w} - T_\text{b}]\Omega \qquad [\text{ML}^2\text{T}^3 \ \text{J/sec}] \qquad (6)$$

Consistent with this introduction, a computer program described in Methods and Supplementary Text was used to develop the theory, giving results consistent with equations (4 -6) described in the following section.

## 2. Results

### 2.1 *Power estimates for simulated commercial wind turbines*

Using available data on blade lengths and estimated chords a set of power outputs, assuming a triangular blade without twist with an angle of wind incidence ($\theta$) 55$^\text{o}$ and tip speed ratio ($\lambda$) as given in Table 1. No attempt has been made in these estimates to optimize aerodynamics, with the windward and leeward surfaces considered as flat and fully elastic. However, it is anticipated that attention to the aerodynamics would produce some marginal effects on power output.

**Table 1. Power outputs from simulated commercial wind turbines**

| Brand | TSR ($\lambda$) $L\Omega/V_\text{w}$ | Wind m/sec | Chord  C meters | Wind torque (Tw) MNm | Back torque (Tb) MNm | Power MW |
|---|---|---|---|---|---|---|
| Vevor 400 | 3 | 15 | 0.112 | 0.0000071229 | 0.00000006913 | 0.0006104 |
| GE 1.5MW | 8 | 15 | 3.025 | 0.61191 | 0.11339 | 1.54380 |
| Nordex N60 | 8 | 15 | 3.250 | 0.41318 | 0.07557 | 1.35043 |
| Nordex N131 | 9 | 15 | 3.500 | 3.46481 | 0.76443 | 4.55715 |
| GE Haliade-X | 10 | 15 | 8.000 | 11.41320 | 2.72240 | 13.0362 |
| | | | | Length (m) | Chord width (m) | Blade area (m$^2$) |
| Vevor 500 mW | | | | 0.520+0.10 hub | 0.023 -0.115 | 0.105 |
| GE 1.5MW | | | | 38.75 + | 0.10-3.025 | 183.0 |
| Nordex N60 | | | | 30 +1.0 | 0.25-3.25 | 120.0 |
| Nordex N131 | 3900 | | | 63+2.5 | 0.25-5.0 | 487.5 |
| GE Haliade-X | 12 MW | | | 100+ 8.5 stalk | 0.5-8.00 | 1290.0 |

$\theta = 55^\text{o}$

Only in the case of the Vevor and Nordex N60 was performance data available to the authors, with 0.400 kW at 12 m per sec wind speed and 1.13 MW at 15 m per sec being claimed. Of several large turbines examined, the Nordex claims were found to be the most modest with claims of 1.3 and 3.9 MW available in advertising material; several of the larger turbines about to be commissioned for marine settings seemed too optimistic, particularly when the rating was conducted as sometimes claimed with wind speed close to 10 meters per sec. We suggest that advertised successful field trials over 24 hours may have been conducted with wind speeds greater than the rating speed.

A requirement for the radial action model was that it should be fully scalable with size. Figure 3 shows power results calculated for a wind turbine fitted with 52 cm blades, equal to those of the Vevor commercial model we purchased. Vevor claim power output of the order shown (Supplementary Text).



These results were obtained for triangular blades of average chord width of 6.50 cm, tapering from
2.30 cm at the tip 62.0 cm from the hub to 11.50 cm at the base, supported on a 10 cm stalk to the hub.

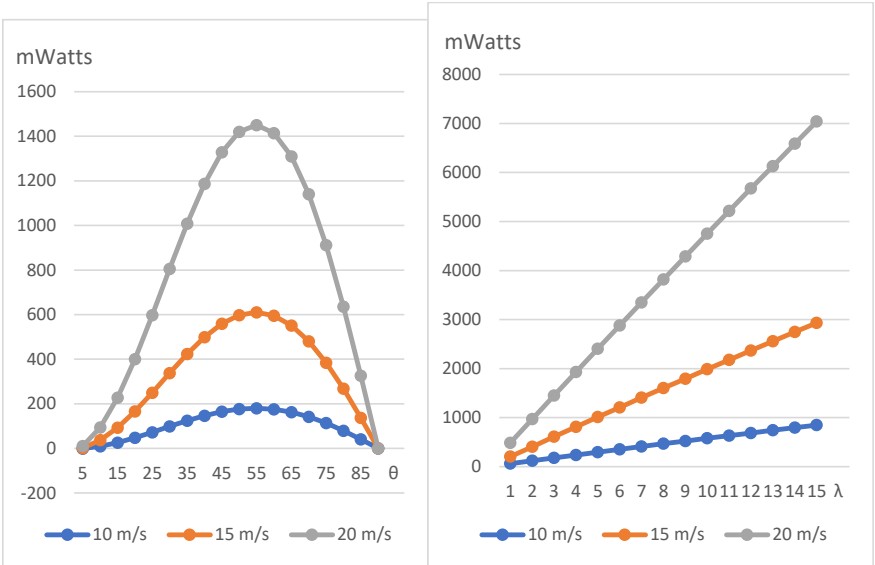

**Figure 3.** Radial action power output in milli-watts estimated for the Vevor 52 cm turbine showing optima for
55-60° wind incidence θ and increasing power with tip-speed ratio λ at all wind speeds. Data for torques Tw and
Tb are given in Table S1 and S2 in Supplementary Text.

Unlike the larger GE 1.5MW and Haliade-X 12MW turbines , the back torque ($T_b$) was negligible
Table 1, reflecting the shorter radius and the $R^2$ factor involved. For the Vevor turbine the theoretical
tip speed ratio (λ) shown in Figure 3 above 4 is excessive, exceeding the limit of 800 rpm by the
manufacturer's guarantee. The program advanced chord width by 0.1769 cm for each iteration. When
data for the Vevor turbine were employed for a rating wind speed of 12 meters per sec with λ of 3.5, a
power output of 406 Watts was calculated. This result is in good agreement with the published rating
of 400 Watts. This result is consistent with accuracy of the radial action model, but it should be
experimentally tested such as in wind tunnels.

**2.2 *Estimates of predicted power output varying wind speed and tip-speed ratio***
Some representative results included in Supplementary Text using the radial action model varying
wind speed and tip-speed ratio similar to those expected for a General Electric 1.5MW wind turbine
of 83 m diameter are given in Figure 4. The data show how the windward [$T_w$] and leeward [$T_b$] torques
differ in character. The former shows a peak close to 55° diminishing towards 0° and 90°. By contrast,
the leeward torque is maximal near zero degrees angle [θ] of incidence, decreasing slowly with the
true angle of incidence to zero at 90°. Assuming a constant angular velocity, the potential power output
shown in the figure has a similar form to the windward torque. The curves assume a given tip-speed
ratio λ and rotational velocity. However, where there is no net torque ($T_w$-$T_b$) the rotor will not
commence or will stall at that angle of incidence. At a wind speed of 10 m per sec, the optimum value





of λ was 8, but when wind speed is 20 m per sec, the optimum ratio was even greater than 20, a value
challenging to the strength of materials, since the tip speed would then be 400 meters per second,
nearing the speed of sound in air.

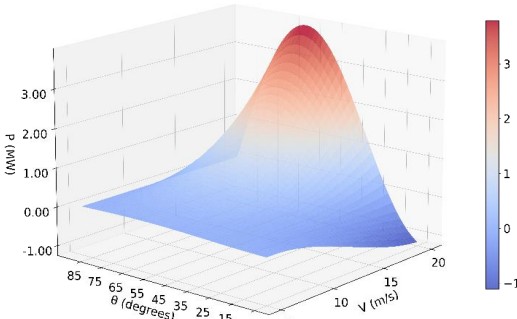

**Figure 4.** Power simulation of 1.5MW turbine using the Wind turbine program A tip speed ratio of 8
was used for four wind speeds of 5, 10 15 and 20 m/sec used for the power diagrams, varying the
angle of wind incidence θ (lower panel). Data for varying tip-speed ratios are given in
Supplementary Text Figure S3.
**2.3 *Comparing kinetic and vortical energies in wind***
From action mechanics we have proposed (Kennedy and Hodzic, 2021a, 2021b) that air in anticyclones
and cyclones, subject to the inertial Coriolis effect, possesses a higher degree of freedom of action
superior to the accepted vibrational, rotational and translational degrees of freedom. This increased
source of entropy can be estimated from its vortical action, capable of magnifying the heat capacity of
air, depending on the radius of action and the vortical frequency or wind speed ($v=R\omega$). By comparison,
the kinetic energy of vortical motion has only a small fraction of the same energy capacity. This is a
testable hypothesis since it predicts that detectable thermal energy will be released as radiant heat from
the cascade of turbulent conditions. Furthermore, colliding air masses must also generate radiant heat
as laminar flow is replaced by turbulence. We consider it is normal function of anticyclones that they
should release radiant heat by friction with the surface (Kennedy and Hodzic, 2021a), an important
natural process transferring heat from the Hadley circulation of tropical air towards the poles. Too
much interference with such natural energy flows could lead to intrusion of colder polar air; this may
already be occurring in the polar vortices being experienced in both hemispheres.






We had originally assumed from our analysis of the Gibbs field in the Carnot cycle that vortical
entropic energy filling a higher degree of freedom of vortical action in anticyclones and cyclones could
be obtained using ambient temperature $T$ expressed as Kelvin. For 3-dimensional translation of
molecules like argon or nitrogen $mv^2$ is equal to $3kT$. For a 3-dimensional velocity $v$, the temperature
($T$) must equal $mv^2/3k$, where the translational velocity ($v$) is 3-dimensional for each gas molecule. By
contrast taking a 1-dimensional velocity ($v = r\omega$) as in horizontal wind in an anticyclone, we conclude
the equivalent temperature ($\tau$) must equal $mv^2/k$ where $v$ is wind speed. With the assumption that
temperature is a statistical version of torque as given in our action revision of the Carnot cycle, the
calculation of mean value negative Gibbs energy must involve the following formula, replacing
ambient temperature $T$ for air molecules with $\tau$ specific to the Gibbs field of wind velocity. Then for
an air molecule with wind velocity of $v$, the virtual wind temperature ($\tau$) must equal $mv^2/k$.

$$-g_{vor} = mv^2\ln[n_{vor}] = s_{vor}\tau \qquad (7)$$

Table 2 provides details of data for vortical entropic energy for a cubic meter of air as wind 1000 km
from the center of an anticyclone. The greater magnitude of the vortical component indicates that wind
power is not so primary function of kinetic energy but more of the vortical energy of the Gibbs field
exerting torques, supporting the motion of the molecules. Calculation of vortical action ($@_{vor}$) and
entropic energy is shown in equation (15), where $R$ is the radial distance to the center of the cyclonic
structure and $R\omega$ the steady windspeed at $R$ from the center.

$$@_{vor} = mR^2\omega; \quad mR^2\omega/\hbar = n_{vor}; \quad \text{Vortical energy per molecule} = mv^2\ln[n_{vor}] \qquad (8)$$


**Table 2. Vortical energy properties for GE 1.5 MW wind turbine**

| Wind speed (m sec$^{-1}$) | Vortical action ($@_v$)/molecule [J.sec, x10$^{19}$] | Quantum number $n_{vor}$ x10$^{-15}$ | 1-D torque $mv^2$/molecule x10$^{24}$ | Vortical energy /molecule [($mv^2$)ln($n_{vor}$), x10$^{23}$ J] | Vortical energy [J/m$^3$] | Kinetic energy [J/m$^3$] | Ratio |
|---|---|---|---|---|---|---|---|
| 5.0 | 2.4215 | 2.29615 | 1.2108 | 4.2826 | 1083.251 | 15.313 | 70.741 |
| 10.0 | 4.8430 | 4.59230 | 4.8430 | 17.4654 | 4417.740 | 61.250 | 72.126 |
| 15.0 | 7.2645 | 6.88845 | 10.8968 | 39.7391 | 10051.703 | 137.813 | 72.937 |
| 20.0 | 9.6860 | 9.18975 | 19.372 | 71.2054 | 18010.864 | 245.000 | 73.514 |

Radius = 1000 km


The relative vortical action and quantum state ($n_{vor}$) are proportional to wind speed but the vortical
energy and quantum field pressure (field energy/unit volume) are logarithmic functions of the action
as quantum numbers. The mean quantum size is exceedingly small and decreases with wind speed,
most of the work or quantum pressure driving the motion of anticyclones (or cyclones) being acquired
at lower temperature and wind speed. Given that air at 288.15 K and 1 atm pressure contains
$2.5294$x$10^{25}$ molecules of air per cubic meter, the vortical field energy of air is 10.052 kJ per cubic
meter at a wind speed of 15 meters per sec (Table 2). For this calculation we assumed a mean mass of
29 Daltons for air molecules to estimate action.






For comparison, Table 3 gives results for associated wind kinetic energy with various wind speeds,
showing data corrected for the Betz limit of 0.593 for the maximum power said to be extractable.
Compared to the rate of kinetic energy passing into the area swept by the blades, the radial action
prediction of power is always less, so if a mechanism to harvest were available, this would be
sufficient. However, when the kinetic energy in the wind actually impacting on the blades is estimated,
this is insufficient to explain the predicted power output. By contrast, when the vortical entropic energy
is estimated (Kennedy et al., 2021) impacting the blades is compared, this is greater than either kinetic
energy and exceeds the actual power output estimated by about six times, almost an order of magnitude
greater.


**Table 3. Kinetic and vortical energy impacting a wind turbine similar to GE 1.5MW**

| Wind speed (m/sec) | Kinetic energy per sec 83 m diam. (Betz) (J) | Kinetic energy /blade-area/sec blades | Vortical pressure (J/m$^3$) (blade area x $v$) | Vortical power (Watts, J/sec) estimated for blade area | Power estimated by radial action model (Watts) |
|---|---|---|---|---|---|
| | | | | | At λ=9, θ=55º |
| 5.0 | 2.1670x10$^5$ | 8.4066x10$^3$ | 0.361069x103 | 0.33038x10$^6$ | 0.031168x10$^6$ |
| 10.0 | 1.7336x10$^6$ | 6.7253x10$^4$ | 1.47258x10$^3$ | 2.69482x10$^6$ | 0.40541x10$^6$ |
| 15.0 | 5.8509x10$^6$ | 2.2698x10$^5$ | 3.35055x10$^3$ | 9.19727x10$^6$ | 1.54381x10$^6$ |
| 20.0 | 1.3869x10$^7$ | 5.3802x10$^5$ | 6.00353x10$^3$ | 21.9729x10$^6$ | 3.86798x10$^6$ |



It is predicted that some of the even larger turbines planned for ocean platforms may not achieve the
performance anticipated. Given that the back torque is a function of the third power of the blade length
when integrated, whereas the windward torque is a function of the blade length squared, a decrease in
performance with increasing blade length is expected. These values were all calculated with chord
length diminishing from the hub to a small fraction of the maximum width, reducing this negative
effect. No claim is made that the estimates in Table 3 are accurate. Chord widths are confidential and
have been estimated from photographs. No account has been taken of the rate of twisting of the blades.
Such reduction in the pitch near the tip selectively reduces the back torque.

*Relationship of vortical entropic energy and Gibbs field to the governing equation of fluid motion*
The governing equations of fluid motion as formulated by Bernoulli, Laplace and others proposed no
such reversible heat-work process for vortices, except absorption and release of heat depending on
whether air is expanding or being compressed as in adiabatic processes. For streamlines as in a laminar
wind flow, the Bernoulli equation (12) relates kinetic energy ($\rho v^2/2$), the static pressure energy $P$
($\Sigma mv^2/3=pV$) and gravitational potential energy, regarded overall in steady flow as constant.

$$\rho v^2/2 + P + \rho gh = K \qquad\qquad (9)$$

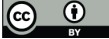



The equation is also the basis of the theory that the pressure on the longer profile of an airfoil will be
lower, given that $\rho v^2/2 + P$ should be constant. The greater velocity required for air flow with a longer
path requires that the pressure $P$ must fall, reducing the downward force. Despite widespread
acceptance of this theory, clear evidence for confirmation is difficult to find.

The vortical entropic energy first proposed in (Kennedy and Hodzic, 2021a) as $S_{vort}T$ must be added
to the total heat content indicated by the Clausius entropy of air (3,4), with the vortical component able
to be released in compressive or turbulent frictional processes. The same categories of energy
transformation are also observed in the Carnot cycle, varying during the isothermal and adiabatic
stages. We have extended this hypothesis (Kennedy and Hodzic, 2021b) to show how this internal
work that Clausius named the *ergal* amounts to a decrease in the Gibbs energy in equation (13). To
the extent that the vortical motion provides an additional degree of freedom for energy storage at a
larger scale, another term needs to be added to the Bernoulli equation.

$$\rho v^2/2 + P + \rho gh + S_{vort}T = K + Nk\tau ln[@_{vort}/\hbar] \qquad (10)$$

This can be thought of as a radial form of quantum state or potential energy, capable of being released
in defined meteorological conditions. We refer to this resonant energy field as the Gibbs field (-$G$) and
it comprises an addition to Clausius' ergal as a work process internal to the atmosphere.

### 2.4 Evidence of vortical heat produced by wind farms
In a previous paper (Kennedy et al., 2019) we showed that surface air heated from absolute zero to
298.15 K needs 2.4 MJ per cubic meter, including its kinetic energy. This implies that the vortical
energy of air in laminar flow at 1000 km of 15 meters per sec from the center of an active anticyclone
contains about 0.42% more wind-reversible field energy than at its non-rotating center. The data in
Tables 3 and 4 of possible heat capacities in the wind suggests an alternative explanation. This source
would be the release of latent heat in wind of vortical entropic energy, a result of turbulence caused by
turbines. The concept of vortical entropy was advanced by as a new class of potential risk to be
considered in climate change. The kinetic energy of wind of 10 m per sec is only 61 J per cubic meter.


**Table 4.  Potential for heat release from turbulence caused by 1.5 MW wind turbines**

| Turbulent process | Vortical energy J/m³ | Heat released J/turbine/sec | Heat released wind farm 70 GE units J/sec | Volume air heated 3 ºC, m³ per sec | Height of warmer air moved 10 m/sec |
|---|---|---|---|---|---|
| Difference | 13593.26 | 5.4370x10⁷ | 3.8059x10⁹ | 1.036x10⁶ | 103.6 m |
| 20 declined to 10 m/sec | | 200 m² blades, 20 m/sec | | $C_p$ 1.225x10³/m³ | 1 km wide front |



Should the laminar flow be impeded by surface roughness, causing turbulence, some of the vortical
energy will be released, warming the surroundings.  For example, as detailed in Table 4, if wind speed
of 20 m per sec is effectively reduced by turbulence to a speed of 10 meters per sec, some 54  MW of
vortical turbulent heat is predicted to be released from air impacting the blades, heating the surrounding



molecules moving downwind. Given a heat capacity of 1.225 kJ per cubic meter for air, this is
sufficient to heat 6,122 cubic meters of air 1 degree Celsius. A windfarm 1 km wide generating 100
MW of power from 70 GE 1.5MW turbines is predicted to release 3800 MW of heat, moving 3.1
million cubic meters of air 10 meters downwind a second, raising air temperature 3 degrees Celsius in
a column about 100 m high. This prediction can readily be directly tested but is consistent with
published observations.

**2.5 *Environmental effects of heat production***
Given the prediction of significant heat production in section 5.1, the environmental effects of wind
farms should be of concern. In particular, their potential effect on evapotranspiration downwind as a
result of turbulence should be considered. Application of the Penman-Monteith equation is the usual
method to model evapotranspiration, including evaporation from soil or water surfaces as well as
transpiration of water used by plants to absorb nutrients, maintain plant turgor and provide water for
photosynthesis. Despite its importance for plant growth in the assimilation of carbon dioxide, the actual
consumption of water for plant growth is far less than that transpired. The inputs required are daily
mean temperature, wind speed, relative humidity and solar radiation. To assist investigation of causes
and effects for these events affecting bushfire risk, we are employing the Penman-Monteith equation
(*UFlorida* 2020 *AE459*), with data potentially of use from the MODIS satellite.

In equation (11) for evapotranspiration (*ET*), factors $R$n and $G$ indicate solar radiation and local
absorption of heat into the soil, $\rho_a$ represents atmospheric density, $C_p$ the heat capacity of air, $e_s^o$ mean
saturated vapour pressure (kPa), $r_{av}$ bulk surface aerodynamic resistance for water vapor, $e_s$ mean daily
ambient vapor pressure (kPa) and $r_s$ the canopy surface resistance (s m$^{-1}$).

$$ET_{sz} = \frac{[\Delta(Rn-G)]+[86{,}400\frac{\rho_a C_p}{r_{av}}(e_s^o - e^a)]}{(\Delta + \gamma(1+\frac{r_s}{r_{av}}))} \qquad (11)$$


Wind speed *u* is also included in the numerator. The main drivers of evapotranspiration are heat from
solar radiation, plant growth, environmental conditions of temperature and relative humidity as well
transport away in air. More important than wind speed, turbulence has now been shown to significantly
increase evaporation, as eddy diffusion lengthens the trajectory for water vapor molecules. Since
terrestrial wind farms are usually placed in rural areas, we are applying this model to test our prediction
that they may contribute to dehydration of the landscape downwind from turbines, increasing fire risk.
We will discuss how these proposals may be tested experimentally, including by observations from
the MODIS satellite.

To determine the potential effects of wind turbines on evapotranspiration, we calculated
evapotranspiration at a range of windspeeds, and then recalculated it with a 1 degree C increase in
temperature assumed to be the result of wind turbines (Table 6). A 1 degree increase in temperature
was used as we found from that turbulence caused in the wind at 20 m per sec by wind turbine blades
effectively reducing laminar speed to 10 m per sec which could release enough heat to raise the
temperature downwind in a swath of 100 m wide and 250 m high more than 50 km downwind by 1





degree Celsius. We used the Penman-Monteith equation for the 5th February at -30.39 latitude, 275 m
elevation and assumed effective daylength of 9.25 hours.
**Table 5.** Predictions regarding heat production and evapotranspiration for 1 $^\circ$C on wind farms

| Wind speed (m sec$^{-1}$) | Evapotranspiration (no wind farm) | Evapotranspiration (with wind farm) | Delta ET (mm day$^{-1}$) |
|---|---|---|---|
| 5.0 | 7.31 mm day-1 | 7.52 | 0.21 |
| 10.0 | 10.18 | 10.52 | 0.34 |
| 15.0 | 12.02 | 12.46 | 0.44 |
| 20.0 | 13.30 | 13.82 | 0.52 |
| 25.0 | 14.24 | 14.82 | 0.58 |

The FAO version of the Penman-Monteith equation was applied using the python module ETo, and
only the values described here were altered. For both temperature and relative humidity, minimum
and maximum values were 20 and 30 C, and 25 and 84%. Evapotranspiration rates calculated without
heat input from turbulence induced by the wind turbines are compared with rates corrected with a 1
degree increase in minimum and maximum temperatures, with all else unchanged. The
evapotranspiration is between 0.21 mm/day at 5 m/s to 0.58 mm/day at 25 m/s. These predictions, and
the short- and long-term effects on the dryness of soil and plants, need to be tested experimentally.
Clearly, at elevated wind speeds over multiple consecutive warm to hot days, the additional quantity
of moisture removed from the soil and vegetation due to wind turbines has the potential to be
substantial. Our calculation in Table 6 has taken no account of the downwind turbulence that may
increase evapotranspiration significantly (Cleugh, 1998; Navaz et al., 2008).
If these results are found in rural landscapes where wind farms are located, they would indicate
increased risk with respect to optimum agricultural or pastoral productivity. However, in some cases
increased temperature and reduced water-holding capacity of air may be beneficial for plant growth,
particularly in environments with ample water supplies. By contrast, in drought prone conditions like
most of Australia, negative effects on productivity and increased fire risk can be assumed. There is a
need for these factors to be assessed wherever wind farms are developed.
494       3. **Discussion**
The radial action model for wind turbines resulted from the search for an imaginative hypothesis that
could be tested, on the lines recommended by the late Karl Popper. The initial computer program
described in Methods was used to develop the model by a reiterative process of trial and error.
Equations (4) and (5) became available only after this process was complete when results and
experience with real wind turbines were found in agreement. It is doubtful if this result could have
been achieved with a higher-level computer language. These equations now form the basis for
presenting the model in other computer codes, like Mathematica and Python available in
Supplementary Text.





The calculation of wind power to the cube of wind speed shown in Equations 1 and 2 considers the
rate at which kinetic energy of air flows through the circular profile area traced by the tips of the
rotors. However, only a fraction of this air can be intercepted by the blades, despite appeal to
concepts like solidity with respect to the air. Particularly in the larger modern turbines most of the air
flow must pass through unimpeded, given that the blades normally represent only some 3-4% of the
area of the rotor circle. Thus, it is likely that no more than 5% of the air volume is initially made
turbulent, effectively tracing triplicate rotary spirals of turbulent air downwind, balancing the work
done on the turbine rotors for transmission into the dynamo.

The leeward torque absent from BEM theory performs work normal to the air flowing into the cavity
behind the blade, up to about 25% of the power generated (Table 1). Additional release of radiant heat
by turbulence is predicted to be a feature of the operation of wind turbines, possibly more than five
times power generation if the vortical energy hypothesis is confirmed. We hypothesize a significant
warming effect downwind that may also increase evaporation, caused by temperature increase and
turbulent surface interaction with vegetation and soil surfaces. This prediction should be tested for
quantification, to be included in productivity models or estimates of fire risk, as a matter of due
diligence.

**3.1 *Reconciliation of radial action and blade element momentum models***
The theoretical success of this model calls for some comparison with the blade element momentum
(BEM) model based on airfoil and Bernoulli fluid motion equations.  In Figure 5(a) aspects of the two
different approaches are given on one diagram, highlighting some differences. The radial approach we
introduce involves two classes of action, quantifying inertial impulses of momentum at each radius of
the rotor blade surfaces, one windward and the other leeward. In the figure, the pitched blades are
regarded as rotating normally to the true wind direction with the turbines facing the wind turning
anticlockwise. Air in inertial motion between the rotors is unimpeded with the blades reflecting
windward impulses at the true angle of incidence ($\theta$). Only a small proportion of the air stream will
impact the blades depending on the proportional area of the circle with radius *L*. If stationary,
obstruction by the blade must create a region of low pressure to its rear. Once a steady state of rotation
$\Omega$ is reached, the blade still obstructs air flow as before but its rear surface impacts resisting air normal
to the blade's motion, deflecting it with the action impulses a function of the radial speed of rotation
($R\Omega$). For most of the blade except near the hub, the pressure at the rear of the rotating blade must be
increased above that in the wind. However, there is no diminution of pressure exerted on the windward
surface because fresh air continually occupies this space. Except near the hub, air at the rear of the
blade is strongly compressed so that no drag in the same direction as the wind is possible, turbulence
caused by diminished pressure.

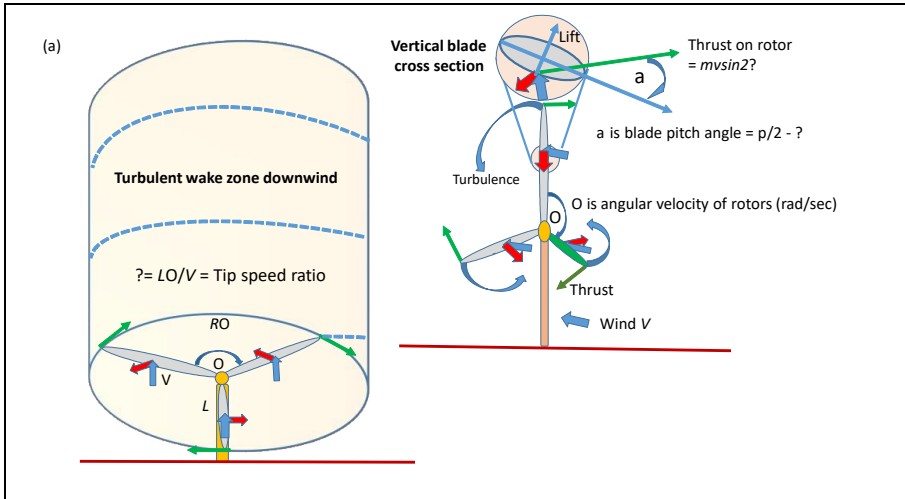


**Figure 5. (a)** Plan and elevation of wind turbine showing clockwise motion with parallel pitching of all three
blades $(\pi/2 - \theta)$ to the plane of rotation with rotor thrust normal to the wind stream. For a tip speed ratio $(\lambda)$
equal to 12 for wind velocity $(V)$ of 10 m/sec, the rotor tips travel 120 meters while the wind advances 10 meters.
Lower: Comparison of blade element momentum (BEM) and radial action theory (RA). BEM subsumes the
reverse or back torque $(T_B)$ as an effect giving relative wind velocity $(W)$. $T_w$ is derived by integrating the
windward action impulses along the blade to the hub and $T_B$ integrates the backward push action impulses
exerted by the blade.

The windward impact pressure is independent of the radius unless the blade is pitched with length,
varying by sin θ, affecting the volume of air impacting the blades per sec. While the intensity of
impulses is constant for a blade surface with a constant pitch, the rate of action or torque varies with
radius, requiring integration for the turbine. By contrast, the impulses produced by the leeward surface
of the blades vary with the radius squared, once for the surface area of air impacted normal to the
rotation per sec and once more for the radial variation in momentum As action impulses.



We have no intention to describe BEM theory (Schubel and Crossley, 2012) in detail. However, in
using a Bernoulli approach to airfoil theory BEM considers lift normal to the blade and drag directed
horizontally along the surface in the wind direction as the two main forces operating. To determine
relative wind velocity the variable speed of an observation point on the rotating blade is taken. If near
the hub, there is little change of direction (θ- β) but near the tip, the apparent angle of incidence
diminishes. In Figure 5(a), the pitch angle to the chord line of the blade is 90-θ degrees, becoming zero
when the true wind is normal to the blade, with no power to cause rotation. Alternatively, the pitch
may be increased to 90°, when the back torque, while the blade is still spinning, will exceed the almost
zero windward torque. Many of the performance studies on horizontal turbines have been performed
with 'frozen' rotors (9), experiments conducted with variable speeds in wind tunnels enabling
theoretical power to be calculated. However, BEM avoids consideration of back torque that impedes
rotation.

Figure 5(b) shows a simulated wind turbine with 40 meter blades similar to the General Electric
(GE1.5) model, generating an average of some 1.5MW of power while rotating clockwise. Note that
by contrast, Figures 1 and 2 would produce a counter-clockwise rotation observed from the windward
direction. The blades represent about 3-4% of the area swept. While most of the air in the wind can
pass between the blades unimpeded, some 10% of the air flow could be subjected to turbulent
conditions. This suggests there is no requirement for a significant decrease in downstream windspeed,
as is required using the BEM kinetic energy model because of the significant decrease in kinetic energy
required. We argued in Section 5 that a lateral source of vortical potential energy sustains the kinetic
energy, except in the turbulent volume where thermal energy will be released.  This prediction of radial
action mechanics can readily be tested.

We emphasize that the main purpose of this article is to develop a testable hypothesis explaining the
maximum power in an ideal wind turbine, assuming the output is linked reversibly to a work process
such as electricity generation. This purpose is similar to Sadi Carnot's proposal that the main purpose
of his heat engine cycle was to describe the most efficient cycle. However, the environmental effect of
turbulence must be examined as it is a key consequence of the radial action hypothesis.

**3.2 *Wind turbine blade design, twist and other modulations in rotors***
The BEM theory justifies the twisting of the blade, reducing the pitch towards zero degrees
approaching the tip (Schubel and Crossley, 2012). A twist is also justified in radial action diminishing
the back torque proportional to $R^2$ nearer the tip whereas the windward torque is no more per *unit* area
nearer the hub. This property could easily be introduced into the radial action model by varying the
relative angle of inclination towards the tip. Further airfoil refinements commonly engineered into the
rotors can easily be incorporated.  These are considered to minimize frictional effects on turbines,
making an independent contribution to the efficiency of power output. Incorporating design features
that are responsive to wind speed and other factors may optimize this process, which can be confirmed
empirically.

This new understanding of power generation from radial action theory accepts the value of research
on optimizing blade design. Factors such as variation in thickness and twist of the chord pitch will still
provide advantages in power output if correctly analyzed. Schubel and Crossley (2012) have provided





a detailed review of the current state of the art of blade design, highlighting efficiency to be gained
from design principles based on blade shape, airfoil properties, optimal attack angles using relative
wind speed and gravitational and inertial properties. With suitable corrections offered by the radial
action approach to wind force, lift and thrust factors and generation of action most of this theory can
be remodeled quite easily.

The Betz limit is considered to exert an effect on BEM theory but has no place in radial action
mechanics.  Only a small proportion of the available vortical energy is consumed, but a different kind
of limit emerges in the competition between windward and leeward torques, with the latter becoming
more significant as the length of turbine blades increases.

**3.3 *Heat production from turbulence***
In developing turbulence, the largest scale eddies nearer laminar flow are regarded as containing most
of the kinetic energy, whereas smaller eddies are responsible for the viscous dissipation of turbulence
kinetic energy. Kolmogorov described by Frisch (1995) hypothesized that the intermediate range of
length scales could be statistically isotropic, and that a temporary form of equilibrium would depend
on the rate at which kinetic energy is dissipated at the smaller scales. Dissipation is regarded as
the frictional conversion  of mechanical  energy to thermal  energy,  effectively  radiation,  raising
temperature. In vortical action theory, the kinetic energy is regarded as always complemented by the
Gibbs field vortical energy and the dissipation process loses kinetic energy, a result and coincident
with  the  loss  of  the  field  energy.  The  dissipation  rate  may  be  written  down  in  terms  of  the
fluctuating  rates  of  strain  in  the  turbulent  flow  and  the  fluid's  kinematic  viscosity,  $v$,  that  has
dimensions of action per unit mass. We suggest that the failure to obtain analytical solutions for
turbulent processes may solved if these complementary forms of energy are considered.

Current practice for wind power makes no provision for heat production other than minimizing friction.
The radial action theory demonstrates that the back torque exerted by turbines is effectively a work-
heat dissipation of wind energy, contributing to its evolution locally at the point of power output.
Depending on the factors controlling efficiency, this heat production can be considered as less but of
the same order as the power take-off as electrical energy.

Of more concern could be additional heat release downwind from turbulence. In Table 4, we provided
estimates, showing that turbulent release is significantly greater in magnitude. While direct heat
production at the turbine is not expected to make a significant difference to air temperature, together
with turbulence, a significant fall in the relative humidity of air passing over vegetation and soil
together with greater surface interaction by turbulent air can be anticipated. The vortical degree of
freedom of motion or action is characterized by its large radius of action, effectively storing latent heat
that can be released as radiation in turbulent conditions. It is known that the kinetic energy in laminar
flow is not retained in the turbulent motion of air or water moving on much shorter radii of declining
scales. Radial action theory predicts this will be the case, the loss of kinetic energy expected as
potential energy we have referred to as Clausius' *ergal* (5) or internal work is released. The kinetic
motion in the system at all scales is sustained by such field or quantum state energy. This is a
consequence of the virial theorem, also explained by Clausius.



The impacts of wind farms on surface air temperatures are well documented. Roy and Traiteur (2012)
claimed that this regional warming of almost 1 °C compared to an adjacent region resulted from
enhanced vertical mixing from turbulence generated by wind turbine rotors. Warmer air from above
the surface, particularly at night, was claimed to be forced to the surface. Harris et al. (2014) showed
"irrefutable night-time warming relative to surrounding areas using observations made from eleven
years of MODIS satellite data with pixel size of 1.1 km$^2$. The same conclusion was reached by Miller
and Keith (2018), showing a significant night-time warming effect at 28 operational US wind farms.
They also concluded that wind's warming could exceed avoided warming from reduced carbon
emissions for more than a century. According to Miller (2020) these effects on warming are detectable
tens of kilometers downwind.
However, the opinion that the warming is a result of overturning temperature inverted air at night is
not convincing. This conclusion is apparently based on an argument that this was the only source of
warmer air considered as available. Vortical heat release now obviates such reasoning. More direct
observations using instruments sensing simultaneously, both upstream and downstream of wind farms,
is required to establish the source of warming, particularly in higher daytime temperatures.
**3.4 *Independent evidence of vortical potential energy as a heat source released by turbulence***
Chakirov and Vagapov (2011) describe a method for direct conversion of wind energy into heat using
a Joule machine. They show that turbulence in a rotating fluid with Reynold's number (*R*e) greater
than 100,000 provides warmth not obtained when the flow is laminar. By insertion of baffles to cause
turbulence in the flow path of water set in rotation in a smooth cylinder using direct wind power, they
demonstrate that needs for room heating in polar regions can be satisfied. It is well known that the
kinetic energy in turbulence is not conserved at lower fractal scales, suggesting that any entropic
energy or ergal is also lost in these processes, where work performed is dissipated as heat by friction.
We have recently discussed such heat-work-heat cycles in an action revision of the Carnot Cycle,
emphasizing the importance of entropic field energy as the negative of the Gibbs energy. The
molecular kinetic energy in such systems is a small fraction of the total non-sensible heat, stored in
quantum state activations of translation, rotation and vibration.
Chervenkov et al. (2013) have shown how the kinetic energy and temperature of polar molecules can
be reduced with a centrifugal force from around 100 K to 1 K, A redistribution of field entropic
potential energy from interior molecules that can be retrieved nearer the center of the centrifuge is
regarded as the cause of the cooling. Geyko and Fisch (2013, 2016) reported measuring reduced
compressibility in a spinning gas where thermal energy is stored in their theory of the piezo-thermal
effect . This extra heat capacity at constant temperature indicates an additional degree of freedom, that
we conclude is vortical, supplementing the well-recognized vibrational, rotational and translational
action as degrees of thermodynamic freedom (3-5).
The widespread failure to recognize the dominance of this nonsensible field energy as real potential
energy (actually, kinetic energy of quanta at light speed [$T = mc^2$, J]) in natural systems, favoring the
sensible kinetic heat indicating the temperature of molecules, has been a critical omission as we
showed in our estimation of the entropy of atmospheric gases and revision of the Carnot cycle. In
effect, potential energy in the atmosphere can be gravitational varying vertically, but it can also be





stored horizontally as vortical energy. These two forms of energy, kinetic and entropic as negative Gibbs energy, are complementary in operation and one must always have one to have the other. With viscous dissipation of energy in storms it is not just the turbulent kinetic energy that is released in a turbulent cascade (Businger and Businger, 2001), but also the vortical entropic energy much larger in magnitude that sustains the kinetic energy. Of course, it is the current enthalpy sustained by the Gibbs field that actually does the physical damage, as we confirmed for work performed in the Carnot cycle (Kennedy and Hodzic, 2021a).

By coupling a Newtonian approach regarding momentum transfer to radial action mechanics (20) regarding torques generated by rates of action impulses as in Gibbs fields we claim we have provided an effective method to estimate maximum power extraction from wind turbines. We claim the following advantages for the new theory, all subject to experimental refutation or confirmation.

- A more effective mathematical model of wind power output. Using the Carnot approach for power of heat engines, the radial action method allows maximum power to be estimated.
- A better template for wind turbine design is also provided, giving an expected closer correspondence between theoretical and practical results.
- A means of optimization of wind power and to minimize heat output is now available. This has the potential to be applied as control theory for managing turbines, either solitary or in wind farms. It is noteworthy that the action mechanics theory developed in this theoretical study suggests that only a small proportion of the potential power of wind in laminar flow is utilized, unlike the BEM theory based on harvesting kinetic energy.
- Environmental protection is also a possible output from using the action model. Ways to manage turbulence to reduce its possible negative effects can be topics for research, seeking to avoid unintended environmental consequences of this burgeoning technology as a prime source of renewable energy.

We emphasize that this paper seeks to present hypotheses that still need rigorous testing. However, for science to advance it is essential that such hypotheses receive due consideration that only prominent publication will allow. This is even more important in an area critical to climate science and the management of climate. The potential of action mechanics to contribute to the science of ecosystems was advanced in 2001 (Kennedy, 2001). A preprint of the paper is also available on the arXiv site (Kennedy et al., 2021.

## 4. Methods

Results reported in the figures and tables are all calculated by exact computation under stated physical conditions and not subject to experimental variation.

### 4.1 *Wind turbine characteristics*

To allow testing of the radial action model for estimation of power outputs, approximate simulations of existing wind turbines were conducted using dimensions shown in Table 1. Blade lengths are advertised, but the maximum chords were estimates made from photographs. No account was taken of pitch values or twisting of the blade.



### 4.2. *Computer Model*

Equations (4) and (5) were determined as results by careful attention to physical dimensions ($ML^2/T^2$), confirmed by the variations in the torques observed in the numerical model, rather than from calculus or basic theory. A key specification was that the radial action wind turbine model should give good results for wide variations in the power outputs predicted, varying with rotor length and blade area from watts to megawatts.

The numerical computer program given in Supplementary Text employed inputs including wind speed ($v$), angle of wind incidence ($\theta$) and tip-speed relative to wind speed, defining a managed angular velocity [$\Omega$, radians sec$^{-1}$] for the turbines. The program was also designed to allow learning by experience. The complexity of current BEM models suggested that a simpler means to determine wind power based on Newtonian physical principles was needed. Our recent papers (3-5) on action mechanics provided physical background to this work.

Elucidation of the governing equations was a result of analyses using numerical methods. The program focused on the calculation of the rate of action impulses [$\delta mvR/\delta t$] as torques, also calculating swept area throughput of concurrent kinetic energy for comparison. Generated on a Windows 10 platform with a capable Texas Instruments SR52 TRS32 emulator, the program is available as Supplementary Text has also been prepared using Mathematica with a Notebook suitable for blades with constant chord C is also available.

To render the program in other systems such as Python and Mathematica, equations (4) and (5) for windward and leeward torques were coded only for relevant sections of the program, without estimates of kinetic energy in swept areas.

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

**Acknowledgments**
We are grateful to the University of Sydney and the Dzemal Bijedic University of Mostar for
infrastructure support.


**Author contributions:**
Conceptualization: IRK, MH, JR
Methodology: IRK, MH, NC
Investigation: IRL, MH, ANC, NA, JR
Supervision: IRK
Writing-original draft: IRK
Writing-review and editing: IRK, MH, ANC, JR
Competing interests: All authors declare they have no competing interests.
Data and materials availability: All original data are available in the supplementary materials.