# Peer review of "A New Way to Estimate Maximum Power from Wind Turbines: Linking 2 Newtonian with Action Mechanics"

_Wind Energy Science, 2022_

## Author Comment (AC2)

```python
################################################################################
**Radial Action example code for three-bladed turbine # Tested on Python 3.8.7 # with the following**
library versions: # matplotlib==3.4.3 # numpy==1.21.2 # scipy==1.7.1 # To run, pass the script
name to python like this: # python radial_action_example.py
################################################################################
import matplotlib import matplotlib.pyplot as plt from matplotlib import cm import numpy as np
import scipy.integrate import scipy.optimize
################################################################################
**Constants**
################################################################################
deg_to_rad = 2.0 * np.pi / 360.0 rad_to_deg = 360.0 / (2 * np.pi) radsec_to_revmin = 60 / (2 * np.pi)
rho = 1.225
################################################################################
**Integrals**
################################################################################
def tw_integrand(r, v, c, theta): return (rho * c * np.sin(theta) * np.sin(2*theta) * np.power(v, 2) * r)
def tb_integrand(r, omega, c, theta): return (np.power(rho, 2/3) * c * np.cos(theta) * np.power(r, 2) *
omega)
################################################################################
**Torque functions**
################################################################################
def torque_windward(v, c_fn, theta_fn, R0, RL, theta0): integrand = lambda r: tw_integrand(r, v,
c_fn(r), theta_fn(theta0, r)) res, _ = scipy.integrate.quad(integrand, R0, RL) return res def
torque_backward(omega, c_fn, theta_fn, R0, RL, theta0): integrand = lambda r: tb_integrand(r,
omega, c_fn(r), theta_fn(theta0, r)) res, _ = scipy.integrate.quad(integrand, R0, RL) return res
################################################################################
**Power and Optimisation**
################################################################################
def power(v, omega, c_fn, theta_fn, R0, RL, theta0): tw = torque_windward(v, c_fn, theta_fn, R0,
RL, theta0) tb = torque_backward(omega, c_fn, theta_fn, R0, RL, theta0) return (tw - tb) * omega
def maximise_power(v, omega, c_fn, theta_fn, R0, RL): ''' Maximises power output over theta0.
Returns the maximum power output of the blade and the theta0 that produces it ''' # scipy
implements only minimisation # the max of a function's values occurs at the min of the negative of
the function's values minimisation_objective = lambda theta0: -power(v, omega, c_fn, theta_fn, R0,
RL, theta0) res = scipy.optimize.minimize_scalar(minimisation_objective) max_power = -res.fun #
negate the minimum to recover the maximum value max_theta = res.x return max_power,
max_theta
################################################################################
**Rectangular blade definition, example turbine configuration**
################################################################################
**blade r0 = 0.00 rL = 38.75 c_fn = lambda r: 1.0 theta_fn = lambda theta0, r: theta0 # turbine L =**
rL λ = 8.0 v = 15.0 Ω = v*λ/L print(f'Ω = {Ω} rad/s') print(f'Ω = {Ω*radsec_to_revmin} rev/min')
################################################################################
**create vectorised forms**
################################################################################
power_vec = np.vectorize(power) torque_windward_vec = np.vectorize(torque_windward)
torque_backward_vec = np.vectorize(torque_backward)
################################################################################
**Example line plot: power, windward and backward torque**
################################################################################
**produce an array of input values for the plots # vary theta0 over [5, 90] degrees, 100 evenly**
spaced points N = 100 θ = np.linspace(5, 90, N) * deg_to_rad # using an array in the theta0
argument, so that the function produces an array result p = power_vec(v, Ω, c_fn, theta_fn, r0, rL,
theta0=θ) tw = torque_windward_vec(v, c_fn, theta_fn, r0, rL, theta0=θ) tb =
torque_backward_vec(Ω, c_fn, theta_fn, r0, rL, theta0=θ) # max power, theta0 for the given
configuration max_p, θ_max_p = maximise_power(v, Ω, c_fn, theta_fn, r0, rL) print(f'θ_max_p:
{θ_max_p*rad_to_deg} deg') print(f'max_p: {3*max_p*1e-6} MW') # plot the power and torques fig,
ax = plt.subplots() ax.plot(θ*rad_to_deg, 3*p*1e-6, linewidth=1.0) ax.plot(θ*rad_to_deg, 3*tw*1e-6,
```

```python
linewidth=1.0) ax.plot(θ*rad_to_deg, 3*tb*1e-6, linewidth=1.0) ax.axvline(θ_max_p*rad_to_deg,
linewidth=0.5, color='k') ax.axhline(3*max_p*1e-6, linewidth=0.5, color='k') ax.set_xlabel(r'Angle of
incidence, $\theta$ (deg)') ax.set_ylabel(r'Power, $p$ (MW)') ax.grid() plt.show()
###############################################################################
**Example surface plot: power as a function of velocity and angle of incidence**
###############################################################################
N = 100 λ = 8 idx_v = np.linspace(5, 20, N) idx_θ = np.linspace(5, 90, N)*deg_to_rad grid_v, grid_θ
= np.meshgrid(idx_v, idx_θ) grid_Ω = grid_v*λ/L lambda8 = power_vec(grid_v, grid_Ω, c_fn,
theta_fn, r0, rL, grid_θ) fig, ax = plt.subplots(subplot_kw={"projection": "3d"}) surf =
ax.plot_surface(grid_v, grid_θ*rad_to_deg, 3*lambda8*1e-6, cmap=cm.coolwarm, alpha=0.8,
linewidth=0, antialiased=True) ax.view_init(15, 220) ax.set_xlabel(r'Wind speed, $v$ (m/s)')
ax.set_ylabel(r'Angle of incidence, $\theta$ (deg)') ax.set_zlabel(r'Power, $p$ (MW)') ax.grid()
plt.show()
```

---

## Author Comment (AC3)

With due respect, we thank Anonymous Referee #2 for the comments related to line numbers on early pages of the manuscript given below.

However, we are disappointed with review #2 since it provides little or no scientific guidance. As our response shows following, not one issue of scientific substance that requires our response is raised by the review. Referee #2 does not engage with the manuscript's substantial content and raises only non-critical matters concerning the style or format of the article.

More critical scientific issues that Referee #2 could have engaged with include the following:

1. Is the application of Newton's experimental method based on the coefficient of restitution for conserving momentum in estimating wind torque and reverse torque valid?  The article assumes that estimating maximum power requires a coefficient of restitution or maximum elasticity of 1.0.

2. This new method provides very credible estimates of wind power for turbine blades varying in length from 0.52 to 100 m and is surely worth examining. The can be experimentally tested, as required for good science.  Readers of this article are invited to participate and wind turbine engineers are well placed to do so, given the open access review in WES.  For example, is the prediction of maximum power at an angle of wind incidence near 60° rather than a pitch of 15° for long blades valid?

3. The application of the novel theory of vortical entropy to wind power, a new degree of freedom in anticyclones and cyclones (Kennedy et al., 2021, *Entropy* 23, 860) can provide new insights for studying turbulence. This thermodynamic (rather than fluid dynamics) theory gains credibility as it explains how the heat of vaporization and condensation of water from the surface of the tropical ocean sustains ongoing power to tropical cyclones. Until now there has been no clear explanation of why only a few percent of the power needed is expressed in the kinetic energy of cyclones, so some form of potential energy must be involved.

4. The hypothesis of vortical entropic energy, a form of horizontal potential energy missing from the Bernoulli theorem, is also able to be experimentally tested with existing and new field data. A consequence of vortical entropy and entropic energy expressed in a Gibbs field is that the vortical wind power can be estimated accurately and can also be subjected to experimental testing, particularly in turbulent conditions.

5. A new environmental risk that needs consideration is the article's prediction that the heat-work involved in increasing entropic energy in anticyclones can be released as frictional heat from turbulence.  Given there is an ongoing debate about heat production and regional warming caused by wind farms, Referee #2 might have considered this issue, even for dismissing it if justified by field data. Due diligence for the location of wind farms should take the quantified prediction of increased dehydration of downwind soils into account, increasing risk of frosts, reduced productivity and conditions possibly worsening wildfires.

In the following list, we respond to the matters provided as a critique by Referee #2.

*II: 16-25. The abstract is far from sufficiently specific. Statements as with the last sentence are rather inappropriate and should be replaced with more specific conclusion and results.*

What could be more specific than the following straightforward statements:

"A new, more accurate way to calculate power output from wind turbines is proposed. This contrasts with current methods regarded as governed by flows of kinetic energy through an area swept by rotating airfoils".

The Referee made no judgement whether this is shown in the paper or not.

The abstract then explains specifically that the new way of "action mechanics measures torques caused by conservation of momentum of impulsive air streams on rotor surfaces at different radii".

"A matter of concern is significant heat production by wind turbines …., mainly from turbulent release of vortical energy.  Use of wind farms as sources of renewable energy may need better practice".

This statement should also have piqued Referee #2's interest, requiring a positive response. Given that we raise the possibility of significant environmental impacts downstream of wind farms, we disagree that our better estimation of optimising power from windfarms is inappropriate.

However, we can agree that more of the content given above could have been included in the abstract if a larger word count was allowed.

*II 28: The introduction includes far too references, so that most statements and given details are not sufficiently verified.*

Given that this article deals with a "new way", it is hardly surprising that it contains few references to current theory.

*II 57: The introduction of "action mechanics" and explanation what is meant by it, comes far too late here.*

Exactly what is meant by a novel action mechanics cannot be explained in a few short sentences. Section 1.1. explains this in the next four pages.

*I 73: Somewhere there should be an introduction of the manuscript's structure.*

We thank Referee #2 for this suggestion for improvement.

*I 80:  It is not clear to me why the authors use square brackets here. Overall, inclusion of maths should be reviewed and revised carefully.*

The  style of this equation follows that of the cited references. No guidance is given of the need for revision of the maths. The Word equation editor is used for all equations, setting the style for the factors in equations.

*I 87: Both Figure 1 and Figure 2 are far too busy and must be revised and improved.*

We claim that the figures are useful as given in showing experimental forces and torques derived from Newtonian theory. However, we thank Referee #2 for suggesting they can be improved.

*I 93: The authors follow a very unconventional structure.*

Obviously, a "new way using action mechanics" will require unique approaches. We cannot avoid this, but would welcome suggestions for a better structure for the article.

*I 239: Also, having a "Methods" section after the "Results" is not common and does not support a well structured manuscript.*

A structure with Methods listed last is recommended by the WES manuscript preparation instructions. However, little would be lost if the section was omitted and its content included in Supplementary Materials.

*I 253: All the details summarized in the table lack suitable references.*

The details requested such as blade chord width are "commercial in confidence" as stated in the text. A statement as a footnote to the table that the details such as blade length are estimates based on measurements from public advertising material collected by the authors should suffice, as indicated in text.

We request that readers of this article can judge whether these concerns expressed by Referee #2 justify not considering the novel and diverse contents outlined above. Not one significant error of either science or of fact is identified in the Referee #2's review.